# Evolution of the Membrane Transport Protein Domain

**DOI:** 10.3390/ijms23158094

**Published:** 2022-07-22

**Authors:** Siarhei A. Dabravolski, Stanislav V. Isayenkov

**Affiliations:** 1Department of Biotechnology Engineering, ORT Braude College, Snunit 51, P.O. Box 78, Karmiel 2161002, Israel; sergedobrowolski@gmail.com; 2Department of Plant Food Products and Biofortification, Institute of Food Biotechnology and Genomics, NAS of Ukraine, Osipovskogo Str. 2a, 04123 Kyiv, Ukraine

**Keywords:** membrane transport protein, MTP, GPR155, PIN/PILS, domain architecture, 3D structural analysis, molecular evolution

## Abstract

Membrane transport proteins are widely present in all living organisms, however, their function, transported substrate, and mechanism of action are unknown. Here we use diverse bioinformatics tools to investigate the evolution of MTPs, analyse domain organisation and loop topology, and study the comparative alignment of modelled 3D structures. Our results suggest a high level of conservancy between MTPs from different taxa on both amino acids and structural levels, which imply some degree of functional similarities. The presence of loop/s of different lengths in various positions suggests tax-on-specific adaptation to transported substrates, intracellular localisation, accessibility for post-translation modifications, and interaction with other proteins. The comparison of modelled structures proposes close relations and a common origin for MTP and Na/H exchanger. Further, a high level of amino acid similarity and identity between archaeal and bacterial MTPs and Na/H exchangers imply conservancy of ion transporting function at least for archaeal and bacterial MTPs.

## 1. Introduction

Transport across a biological membrane is a pivotal process for all kinds of living organisms. This process plays an important role in various physiological and biochemical reactions and pathways, such as nutrition, signalling, ion homeostasis, metabolite deposition, and others. Therefore, the functional and structural study of membrane transport proteins should be a paramount research direction for future studies.

Membrane transport proteins (MTPs) are widely presented in all studied taxa of living organisms. However, the detailed study and functional characterisation of this protein family remain obscure for most life forms. The basic MTP domain architecture comprises 10 transmembrane domains with an overall length of the domain of around 300 amino acids. Currently, the function of MTPs is unknown. Many bacterial and archaeal proteins are annotated as “Transporter”, “AEC (Auxin Efflux Carrier) family transporter”, “Putative permease”, or “Malate permease”. Similarly, fungal proteins are mostly specified as “unknown”, “uncharacterised”, or “AEC family transporter” proteins, however, no studies have been conducted to specify transported substrate or performed function. Some proteins from species of Opisthokonta taxa (Choanoflagellata) and many Metazoan proteins are annotated as “DEP domain-containing protein”, “integral membrane protein GPR155-like”, and “Putative transporter YfdV”.

Originally, Gpr155 (G-protein-coupled receptor 155) was identified in Huntington’s disease patients, where it was reported to be downregulated in the caudate nucleus, suggesting involvement in pathophysiological mechanisms affecting specific brain structures [1]. Further, the Gpr155 dysregulation was confirmed in the R6/2 mouse (model system of Huntington’s disease), where it was dysregulated in the striatum-enriched transcripts [2]. Later, the gene structure and expression profile have been studied in mice, where *GRP155* expression in the forebrain, midbrain, and hindbrain was confirmed. In addition, the predicted protein secondary structure suggested 17 transmembrane domains and a conserved DEP domain at the C-terminus of Variant 1 and Variant 5 proteins [3]. Further studies on the lateral region of the mouse striatum demonstrated a similar expression profile of *GAD1* (GABA-synthesizing enzymes glutamate decarboxylase), cannabinoid receptor type 1, and *GPR155*, suggesting functional cooperation between these proteins in the modulation of sensorimotor and limbic inputs in the striatonigral and striatopallidal pathways [4], which are specialized for control of motor and motivational behaviours [5]. *GPR155* dysregulation is involved in ASDs (autism spectrum disorders) [6] and type 2 diabetes development [7], however, the molecular mechanism and its pathophysiological role remain unclear. Additionally, *GPR155* could have critical functions in the treatment of methamphetamine addiction with Topiramate [8]. Furthermore, GPR155 was confirmed as a crucial biomarker, associated with several types of cancer. *GPR155* downregulation was associated with more aggressive hepatocellular carcinoma phenotypes [9], hematogenous metastasis of gastric cancer [10], and the follicular variant of papillary thyroid carcinoma [11]. However, on the contrary, Gpr155 was upregulated in UVR-induced mouse melanomas [12]. Moreover, the GPR155 I357S mutation was associated with the acquired resistance to chemotherapy in lung cancer patients [13]. Moreover, GPR155 was studied in several other experiments. For example, *GPR155* knockdown in *Drosophila melanogaster* resulted in thickened and ectopic vein phenotype and the development of enlarged wing size. Further investigation showed that *GPR155* negatively regulates BMP (bone morphogenetic protein) during wing development and plays an important role in tissue patterning [14]. Genome-wide association analysis on pigs demonstrated that GPR155 is involved in regulating residual feed intake, a complex trait that might be useful for the genetic selection of more feed-efficient pigs [15].

Plant (Viridiplantae) MTPs are defined as PIN (PIN-FORMED)/PILS (PIN-LIKES) proteins and are the most studied group among all MTPs. The PIN/PILS participate in the transport of plant hormone auxin [16]. Therefore, the functions of this type of transporter protein are crucial for plant development, general architecture, and environmental adaptive responses [17]. Subsequently, the localisation and post-translational modifications of PIN/PILS proteins affect auxin transport and distribution, which have well-studied physiological and developmental consequences [17,18,19]. The evolution of the PIN/PILS proteins was studied only within Viridiplantae taxa [20], with experimental confirmation of auxin-transporting properties for PIN protein derived from green alga *Klebsormidium flaccidum* [21]. However, despite experimental pieces of evidence confirming auxin transport properties for some members of the PIN/PILS family, the substrate specificity of this group of proteins in plants was not screened for other substances. In addition, no experiments are proving/disproving the ability of the PIN/PILS proteins to transport other than auxin bioactive molecules or metabolites. Furthermore, the ability to transport auxin was not shown for MTPs from other taxa (Archaea, bacteria, or Opisthokonta). Similarly, no experiments are proving/disproving the ability of PIN/PILS proteins to transport other than auxin bioactive molecules or metabolites. The experimental data about the functionality and physiological role of MTPs from the other taxa, including fungi, bacteria, archaea, and animals, remain obscure. Therefore, this important research area is still waiting to be substantially developed.

Phylogenetic and structural studies of MTPs have not been carried out yet. However, recent progress in the genome sequencing of the various taxonomic groups significantly clears the way for comparative studies across a much wider variety of species from different taxa. In this paper, we reconstructed 3D structures from the different taxa to perform a comparative analysis. The comprehensive MTP phylogenetic inventory was conducted to evaluate the molecular evolution aspects of this big group of membrane proteins. 3D structure and sequence analysis indicate the common evolutional origin of MTP and Na/H exchanger domains, suggesting the possibility of ion transport or more diverse substrate specificity of MTPs.

## 2. Results

### 2.1. Sequence, Domain Architecture, and Loops Identification

To track the evolutionary origin of the MTP domain, we have searched for the homologous sequences from the UniProt, Pfam, and InterPro databases. Although this family is widely presented in all taxa, we have downloaded only a few representative proteins from diverse taxa. During sequences search, the truncated, partial, and identical sequences were removed.

Interestingly, all analysed archaeal MTP domain-containing proteins are represented by a single-copy gene, which is encoding a single-domain protein of approximately 300 amino acids with 10 TM domains. Similarly, most bacteria taxa (FCB group, Proteobacteria, minor bacteria taxa; minor terrabacteria, Firmicutes; the majority of actinobacteria and cyanobacteria) have similar domain architecture and protein length. Some differences appear in terrabacteria taxa (Mollicutes, Cyanobacteria, and Actinobacteria), where cytoplasmic and non-cytoplasmic loops of different lengths are presented between V–VI and VI–VII, III–IV, and VI–VII, III–IV (Mollicutes), V–VI (Actinobacteria), and III–IV and V–VI (Cyanobacteria) TM domains (Figure 1). Further, in the Eukaryotes (Amoebozoa, Discoba, Haptista, Metamonada, Rhodophyta) the basic domain architecture was the same as for Archaea and bacteria (single-domain 300 aa in length with no loops), some proteins had cytoplasmic loops in different positions: between III–IV, V–VI (Amoebozoa), V–VI, V–VI and VII–VIII (Discoba), V–VI and VII–VIII (Haptista), V–VI (Metamonada and Rhodophyta) TM domains (Figure 1). SAR taxon had some variation in the domain architecture and loops positions. The Rhizaria and Alveolata had only a central cytoplasmic loop presented (between V–VI TM domains), and some MTP domain-containing proteins from Stramenolipes had 2 cytoplasmic loops (V–VI, V–VI, and VII–VIII TM domains) (Figure 1). Additionally, some Alveolata MTP domain-containing proteins had domain architecture usual for the Metazoan taxa–MTP domain, followed by DEP domain. In Opisthokonta taxa, represented by Fungi, Metazoan, and other minor taxa (Choanoflagellata, Filasterea, and Ichthyosporea), diverse domain architecture and loop position/length were observed. Choanoflagellata have MTP domain-containing proteins with unusual non-cytoplasmic loop topology (between VIII–IX TM domains) and DEP domain. Ichthyosporea had only a central cytoplasmic loop (between V–VI TM domains) with no additional domains, whereas Filasterea had two cytoplasmic loops (between V–VI and VII–VIII TM domains). Similarly, Fungi were presented mostly with a central cytoplasmic loop (Ascomycota), or two cytoplasmic loops (between V–VI and VII–VIII TM domains) (Basidiomycota). Further, the Metazoan taxa have basic topology presented with MTP domain, followed by DEP domain (Porifera, Cnidaria, Vertebrata), some of the minor members had loops (central cytoplasmic loop in Protostomia) and two loops in Deuterostomia (central cytoplasmic loop combined with non-cytoplasmic loop between VIII–IX TM domains) (Figure 1). Interestingly, the central loop is always present in Viridiplantae taxa, where the MTP domain-containing proteins are classified based on the length of the loops as PIN (PIN-FORMED) and PILS (PIN-LIKES) proteins (Figure 1).

### 2.2. Phylogenetic Analysis

To understand the evolutionary history of the MTP domain, the phylogenetic tree was inferred with the maximum likelihood method. For most bacteria and archaea, the entire MTP domain-containing proteins were used. For other taxa where the MTP domain was a part of multi-domain proteins, the sequences of the MTP domain were extracted. Loops introduced disturbance in the alignments and were removed during sequence processing. Many plant proteins (with long loops) did not pass the sensitivity filter and did not appear in the final tree (Appendix A). Despite the poor statistical support at deep nodes, we could notice a good separation on clades for high-ranked taxa.

In the used conditions, we define 13 clades with some sub-clades (marked out with a-b letters). The Clade I was formed mostly with Archaea MTP proteins, 3 bacterial proteins of different taxa (Cyanobacteria—*Symploca* sp. SIO2G7, Actinobacteria—*Acidimicrobiales bacterium*, a unique extremely thermophilic bacterium *Thermosipho japonicus*), and eukaryotic organism *Entamoeba histolytica*. Clade II was formed by Terrabacteria (sub-clade IIa with Firmicutes and sub-clade IIb with Cyanobacteria) and proteins from some other bacterial taxa. Although Clade III consists mostly of Terrabacteria (Firmicutes) and some other bacteria, there are also two primitive eukaryotic organisms like an outgroup: fungus *Orbilia oligospora*, known to capture nematodes, and microscopic single-celled flagellated protozoan parasite *Tritrichomonas foetus*. Clade IV has only bacterial MTPs, which mostly belong to FCB group and Proteobacteria taxa. Clade V is divided into two sub-clades, of which clade Va is formed by bacteria (mostly Proteobacteria taxa) and Vb is more diverse and includes different bacterial (Firmicutes, FCB group and Proteobacteria) and simple Eukaryotic organisms (Metamonada taxon—*Trichomonas vaginalis*) and proteins. Interestingly, three bacterial MTP proteins (FCB group—*Chitinivibrio alkaliphilus*, Firmicutes—*Helcococcus kunzii* and *Brevibacillus centrosporus*) and two archaeal (*Crenarchaeota group archaeon* and *Candidatus Korarchaeota archaeon*) act as an outgroup for the Clade V.

The Clade VI is formed entirely by Proteobacterial MTP proteins, and only *Candidatus Nitrohelix vancouverensis* (belongs to Tectomicrobia group) protein is in an outgroup position. Clade VII comprises two sub-clades, sub-clade VIIa is formed by Cyanobacteria branch (four MTP proteins), and a small side branch with Eukaryotic SAR Alveolvata (endosymbiotic dinoflagellates *Symbiodinium microadriaticum*), Alphaproteobacteria (*Varunaivibrio sulfuroxidans*), and Terrabacteria (*Dehalococcoidia bacterium*), with a distant relation to the Archaea *Halarchaeum acidiphilum*. Sub-clade VIIb is less diverse and contains mostly Firmicutes. Similarly, the VIII clade has two sub-clades: VIIIa contains Actinobacterial and Proteobacterial MTP proteins, alongside a primitive animal *Hyalella azteca*, and sub-clade VIIIb is formed mostly by Proteobacteria. The smallest identified Clade IX has a diverse taxonomy set: four Proteobacteria, a primitive Eukaryotic protist *Palpitomonas bilix*, known for its uncertain taxonomic affiliation, and myxozoan endoparasite *Thelohanellus kitauei*.

The remaining clades X-XIII are represented only by Eukaryotes. Clade X consists of three sub-clades: sub-clade Xa has only Viridiplantae species (both basic Viridiplantae and Dicotyledons); sub-clade Xb—only Monocotyledons; and sub-clade Xc—all set of Viridiplantae (basic, Mono- and Dicotyledons). Similarly, the Clade XI is formed mostly by basic Viridiplantae species, for Dicotyledons species (*Brassica cretica*, *Nyssa sinensis*, *Gossypium trilobum*, and *Senna tora*) and only one Monocotyledon (*Spirodela intermedia*). The Clade XII has more diverse organisms: basic Eukaryotes of different taxa (Cryptophyceae—*Guillardia theta*, Apusozoa—*Thecamonas trahens*, Rodophyta—*Gracilariopsis chorda* and *Porphyra umbilicalis*, Haptista—*Emiliania huxleyi*, and Amoeba—*Arcella intermedia*), Viridiplantae (green algae—*Ostreococcus tauri* and *Ostreococcus lucimarinus*) and SAR (Alveolata—*Symbiodinium microadriaticum* and *Symbiodinium natans*, and Stramenopiles—*Florenciella parvula*). The sub-clade XIIb consists of SAR species (both Alveolata and Stramenopiles, *Vitrella brassicaformis*, *Fistulifera solaris*, and *Nannochloropsis salina*, respectively), several Rodophyta species (*Gracilariopsis chorda*, *Rhodosorus marinus*, *Porphyridium purpureum*, and *Timspurckia oligopyrenoides*), Discoba (*Trypanosoma cruzi* and *Leishmania donovani*), and Fungi (several species of both Ascomycota and Basidiomycota). The clade XIII has two sub-clades, where sub-clade XIIIa is formed by Metazoan species from both taxa Deuterostomia and Protostomia, and the sub-clade XIIIb, only by Deuterostomia. Several proteins of diverse taxa were not included in any of the sub-clades of the XIII clade and formed separated branches, parenting the rest of the XIII clade: sponge *Amphimedon queenslandica*, the closest living relatives of the animals from Choanoflagellate taxon—flagellate eukaryotes *Monosiga brevicollis*, biflagellate monades from Cryptophyceae—*Hemiselmis andersenii*, and SAR Alveolate—*Strombidinopsis acuminata*.

Interestingly, three bacterial MTP proteins act as an outgroup for the entire phylogenetic tree, which belongs to the Proteobacteria (*Candidatus Muproteobacteria bacterium*), Terrabacteria (*Anaerolineaceae bacterium* and *Acidimicrobiaceae bacterium*).

### 2.3. Highly Conserved Residues in Transmembrane Domains of MTP Domain-Containing Proteins

To further understand the structure of MTP domain-containing proteins, we analysed their transmembrane domain structure. In general, the overall length of the MTP domain was about 300 (excluding loops) amino acids and comprises 10 transmembrane domains. However, we should notice that TM domain prediction strongly relies on the used software, which could predict other numbers of TM domains with the presence of cryptic peaks corresponding to the “missing” helices. Thus, we conclude that all MTP domains are highly likely to have exactly ten helices. Similarly, depending on the used software, the length of the TM domain could vary by about 2–5 amino acids (shorter/longer).

Through all examined taxa, TM domains I, II, IV, and VI-IX have only 1 highly conservative amino acid, and 2–6 less conservative. Interestingly, TM domain V has only 1, TM X has 3 less conservative amino acids, whereas TM III has no conservative residues at all (Appendix A). Because of the presence and stable position of highly and less conserved amino acids throughout the entire tree of life, we could suggest a certain degree of evolutional stability of the transmembrane domains’ sequences. Furthermore, for every taxon, there are exceptions where the highly conservative amino acid is replaced, whereas the surrounding less conservative amino acids are presented and vice versa. It would also be important to define function-related meaning for such a substitution: is it related to the protein stabilisation within the membrane, transported substrate selectivity, or substrate translocation process? The absence of conservative amino acids in TM domain III, and the limited number of less conservative amino acids in TM domains V and X (1 and 3, respectively) suggests that the exact amino acid composition of these transmembrane helices is rather unimportant for structural stability and performed function. However, further experimental studies are required to confirm those assumptions.

### 2.4. Loop Analysis

The main site of variation in the structure of MTPs from different taxa is the lengths of the loop between TM domains. Therefore, in the next step, we have analysed loop sequences (both cytoplasmic and non-cytoplasmic), which connect transmembrane domains. In general, we could notice that bacterial and Metazoan loops are much shorter compared with loops sequences from other taxa, with the majority of bacterial and Metazoan loop lengths approximately 40–60 amino acids, and Viridiplantae up to 520 amino acids. Additionally, the Cytoplasmic loops are more prevalent compared to non-cytoplasmic. The comparison of loop positions in relation to TM domains shows that the loop’s location between V and VI TM domains is dominant, whereas other TM positions (III–IV, VII–VIII, VIII–IX and VI–VII) are much less abundant, with no loops located between TM I–II, II–III or IX–X.

Interestingly, the BLAST examination of bacterial loop sequences returns only the closest bacterial species, whereas loop sequences derived from Eukaryotic MTP domain (excluding Embryophyta) are unique, and BLAST scans are unsuccessful (data not shown). Similarity, alignments of loop sequences derived from distant taxa are unsuccessful, with no conserved residues between and within bacterial and Eukaryotic taxa (excluding Embryophyta). On the contrary, loop sequences derived from Embryophyta have a high level of similarity and include a set of well-conserved motifs (those motifs were extensively studied [20], therefore, excluded from our analysis).

Finally, independent of the length, assigned TM domain location, and taxa, all loop sequences are specified as intrinsically disordered regions (IDRs) (several representative proteins shown in Appendix A).

### 2.5. Structure Simulation and Analysis

Currently, no solved crystal structure is available for any MTP domain-containing protein homologous. For structure prediction, we used the Robetta server, which applies both a template-based search (if available) and de novo structure prediction [22]. Reconstructed crystal structures were used to define the effect of loops on the structure, and to predict the evolutionary aspects of protein function and mechanism of transport. As an intrinsically disordered region, all loop regions in reconstruct structures had low confidence scores, whereas TM regions had high and very high. Regardless of the loop/s position and their length, all MTP domain-containing proteins with loops were well-aligned to proteins without loops (Figure 2). To conduct alignment, all extra domains (such as DEP domain in animals) were excluded and the predicted model was limited to the MTP domain region. Interestingly, Arabidopsis PIN5 shows the best alignment score among all tested loop-containing MTPs to loop-less control proteins from bacteria, sponges, and humans (Figure 2). In general, the alignment of reconstructed structures with and without loops, representing diverse taxa, suggests a high level of conservancy for the MTP domain-containing proteins between all living organisms and that loop position and length do not affect a protein’s structure.

In the next step, we have implemented a VAST search to find similar structures in MMDB (Molecular Modelling database). Only bacterial and Archaeal MTPs displayed significant similarities to 4BWZ and 4CZ8, which represent sodium proton antiporters from *Thermus thermophilus* and *Pyrococcus abyssi*, respectively, with Na/H exchanger domain (PF00999 or IPR006153). Regardless of the presence of 13 helices, the alignment of modelled structures (MTPs vs. Na/H Exchanger domain) within one organism indicated a high level of similarity only for bacteria and Archaea (Figure 3), whereas alignment scores were not significant for other taxa. Interestingly, MTPs and Na/H exchanger domain belong to one clan (CPA_AT (CL0064)) of transporters. Despite that, we didn’t find structural similarities between arabidopsis PINs/PILs and CHXs, and KEA4 and NHX1 (all containing Na/H Exchanger domain) (data not shown), suggesting the development of substantial functional and structural differences during evolution.

### 2.6. MTP/Na-H Exchanger Comparison

The alignment of amino acid sequences demonstrated a high level of similarity/identity between archaea MTP and Na/H exchangers (Appendix A), thus confirming alignment results of modelled structures with bacterial and archaea Na/H exchangers (Figure 3). The mechanism of action of archaea *Pyrococcus abyssi* Na/H exchangers is well-studied and the main ion-binding sites and coordinating residues are known [23]; we compared those sites in archaea MTP (F8AHV6, I3RCU4, I6UZT4, and Q8U3L6, representing differently-related *Pyrococcus* species and strains).

The main substrate ion-binding site, defined at pH 8, is formed by Glu73, Asp159, and Asp130, whereas Thr129 and Ser155 provide two additional ligands. The second, narrow polar channel next to the cytoplasmic funnel leads to an enclosed polar cavity near Asp93, Thr129, Asn158, and highly conserved pair Glu154/Arg337. The Glu154/Arg337 ion bridge and Thr129 separate the cavity from the narrow polar channel. At pH 4, the structure undergoes some rearrangements, most importantly, near the dimer interface and blocking the second narrow polar channel by the surrounding residues Ile151, Phe355, and Gly359 (Appendix A). Although the main substrate ion-binding site was almost completely missing (except for Ser155), the narrow polar channel had two similar amino acids (Asp158 and Arg337) and matching blocking residues (Phe355 and Gly359).

## 3. Discussion

MTPs are widely presented in all living organisms. Unfortunately, their functional properties are still largely unexplored. Similarly, the evolution was analysed only for the limited number of plant taxa [20]. Our phylogenetic analysis suggests 13 clades with well-distinguished separation by high-ranked taxa. Clades distribution (Appendix A) with low bootstrap support of the deep nodes and the level of similarities on amino acids level (Appendix A) suggest common origin with independent clade-specific specialisation. Interestingly, in several clades, we could find some species representing other taxa, for example, such as a parasite or symbiotic eukaryotic, bacterial, and archaeal species, which could suggest some kind of gene transfer between species of different taxa. Several such cases were identified: (1) Terrabacteria *Acidimicrobiales bacterium* and *Symploca* sp., separated branch with bacteria *Thermosipho japonicus* and eukaryote *Entamoeba histolytica* in Archaea clade I; (2) eukaryotes *Orbilia oligospora* and *Tritrichomonas foetus* in bacterial clade III; (3) eukaryote *Trichomonas vaginalis* in bacterial sub-clade Vb and a separate branch with archaea in bacterial sub-clade Va; (4) eukaryote *Symbiodinium microadriaticum* and archaea *Halarchaeum acidiphilum* in bacterial sub-clade VIIa; (5) eukaryote *Hyalella azteca* in bacterial sub-clade VIIIa; (6) Clade IX, which is represented with diverse bacterial (*Cronobacter malonaticus*, *Frischella perrara*, *Entomomonas moraniae* and *Francisella uliginis*) and eukaryotic (*Thelohanellus kitauei* and *Palpitomonas bilix*) species; (7) a primitive green algae (*Ostreococcus tauri* and *Ostreococcus lucimarinus*) in basic eukaryote sub-clade XIIa; (8) a primitive eukaryote (SAR *Strombidinopsis acuminata*, Cryptophyceae *Hemiselmis andersenii*, Opisthokonta *Monosiga brevicollis*, and Metazoa *Amphimedon queenslandica*) in the advanced Deuterostomia and Protostomia clade XIII. Apparently, those cases require close examination in future studies.

It is important to note that the presence of a loop affects the amino acid alignment quality and, subsequently, the phylogenetic tree. Therefore, the loop region was cut and analysed separately. This is crucial, especially for the proteins with a long central loop, which could be longer than the rest of the protein. Another reason, justifying this approach, results from using a domain architecture predicting tool (such as the NCBI CD-search tool), which defines a loop region as a gap (in case of short loops) or unorganised region (in case of long loops).

Single-domain is the prevalent domain architecture for MTP domain-containing proteins, whereas there are a few other domains, often associated with MTPs (such as the DEP domain in Metazoa). Furthermore, functional MTP comprises structural 10 transmembrane domains (Appendix A) which are the most highly conserved among studied species (except TM3, TM5, and TM10). Because we did not find MTP with loops between TM1-TM2, TM2-TM3, and TM9-TM10, but TM5-TM6 is the prevalent loop position, it would be worthwhile to specify the relation between conserved amino acids in TM domains and loop/s position. Because all identified loops were unique (except Embryophyta) it is also essential to determine how the loop’s position and length are related to MTP function, localisation, and transported substrate specificity. Loops are well-studied in Embryophyta, where there are several conservative regions have been identified [20]. Additionally, the loops region of the plant (specifically, PINs and PILS of *Arabidopsis thaliana*) were shown to contain sites for PTM (post-translational modifications) such as phosphorylation and ubiquitylation, which regulate polar delivery and its efflux activity [reviewed in [16]].

Furthermore, our in silico prediction shows that all loops (independent of length, position, and taxa) are IDP (intrinsically disordered protein) (Appendix A), which suggested their role in protein–protein interaction. Experimentally, the IDP nature of loops was confirmed only in Arabidopsis PIN1 protein, where it is involved in regulating PIN1 trafficking, which resulted in minor phenotype changes, and, possibly, dimerisation and interaction with other proteins [24]. In this regard, it would be interesting to define the role of non-cytoplasmic loops and their possible role in dimerisation for nonsymmetrical loops (located on TM3-TM4, TM6-TM7, TM7-TM8, and TM8-TM9). Therefore, further intensive experimental studies with loop truncations or the construction of chimeric proteins with domains/loops swapping would be promising directions for research. In a particular case, to reveal the functional role of loop regions of MTPs, the exchange of small loops for a big one, or the addition of plant loop/s to the animal MTPs and vice versa, might be useful. In plants, it is also essential to find an effect of loop region translocation to the other than the TM5-TM6 position.

The results of our comparison of modelled MTP structures confirm the close relation and common origin of MTPs from different taxa, defined by phylogenetic tree analysis and alignment of amino acid sequences (Figure 2). In addition, these results suggested that loops (their position, length, and direction (non- or cytoplasmic)) do not affect the general structure of MTPs’ TM domains. Because of the high level of similarities between MTPs from different taxa, it is tempting to speculate that MTPs share a general function as a transporter. However, the role of loop/s in transporting (protein localisation, substrate selection, or regulation by interaction with other ligands) is yet to be found. So far, the only known transported substrate is plant hormone auxin, which was shown in many plant species [reviewed in [17]]. Although the expression of plant PINs and their ability to transport auxin was confirmed in different species (including *Saccharomyces cerevisiae*, *Xenopus laevis*, and human HeLa cells) [19], there are still no data indicating the transport of auxin by some non-plant MTPs. In addition, some plant PINs/PILs are still not characterised well. Perhaps, some of them could exhibit different substrate specificities and transport properties. From the evolutional point of view, analysis of MTPs from other taxa would help to understand the origin of plant hormone auxin, when the simple metabolite indole-3-acetic acid first appeared and acted as a hormone, how it is transported and metabolised in other non-plant organisms, and what signalling pathways evolved for its regulation.

Interestingly, the search for similar 3D structures using the VAST tool returned significant similarity results only for bacterial and archaeal MTPs, which were matched to the Na/H exchanger (Figure 3). Basically, these results confirm the earlier paper [25] where both Na/H exchanger and MTP belong to one clan (CL0064). However, significant similarities between modelled MTPs and Na/H exchangers were found only for bacterial and Archaeal species. Further analysis of functional residues identified in Na/H exchanger with the solved crystal structure (4CZ8 [23]) demonstrated conservancy of some crucial channel-forming residues, supporting our idea of common origin and functional similarities at least between bacterial and archaeal MTPs and Na/H exchangers. However, further research is required to define transported substrate for MTP from different taxa.

Although for Metazoan and Fungal no data is available to connect our results of close evolutional relation between MTPs and Na/H exchanger, several studies on plants demonstrated the tight functional connection between PINs/PILs (plant MTPs) and Na/H exchangers [26,27]. As it was shown on the quadruple knockout *nhx1nhx2nhx3nhx4*, lacking all vacuolar NHX transporters, which had auxin-related phenotypes including reduced root growth and sensitivity to exogenous auxins IAA and NAA, impaired gravitropism and loss of apical dominance. Moreover, in the quadruple nhx1nhx2nhx3nhx4, the abundance of the auxin efflux carrier PIN2, but not PIN1, was significantly reduced at the plasma membrane and was concomitant with an increase in PIN2 labelled intracellular vesicles [28]. Little is known about the role of vacuolar ion (Na^+^/K^+^) transport in auxin homeostasis, whereas NHXs (CPAI) play an important role in modulating auxin homeostasis, most probably, other ion transporters with Na/H Exchanger domains (such as CHX and KEA (CPAII)) would also participate in auxin distribution via PINs/PILs modulation. Both (MTPs and Na/H Exchangers) act as a dimer, thus it is also crucial to check their ability to make heterodimers. In addition, ion homeostasis could affect loop PTMs status or Na/H exchanger could directly interact with MTPs’ loop. Furthermore, it would be interesting to screen PINs/PILs mutant lines on Na^+^(K^+^)/H^+^ exchange activities and the abundance of NHX/CHX/KEA transporters. Perhaps further application of pull-down assays or other methods used to specify the interaction between different proteins may shed the light on this functional and structural aspect of plant MTPs.

Another attractive approach to studying the functional properties of MTPs from different taxa is the heterologous expression of genes encoding MTPs homologs in loss of function mutants of different life forms including bacteria, yeasts, animal cell lines, and plants. This approach may help to reveal some functional and transport properties of MTPs from different taxa. As was mentioned above, bacterial and archaeal MTPs have significant structural similarities with the Na/H exchanger domain, and therefore, the ability to transport ions, in particular, Na^+^ and K^+^, could be conserved at least for the primitive living forms. On the contrary, the transported substrate specificity of MTPs from fungi, bacteria, archaea, and animals is still completely unknown.

## 4. Materials and Methods

### 4.1. Sequences Retrieval

The proteins containing the membrane transport protein (MTP) (Pfam PF03547, InterPro IPR004776, YfdV COG0679) were downloaded from NCBI (http://www.ncbi.nlm.nih.gov/ (accessed on May–June 2022)), UniProt (http://www.uniprot.org (accessed on May–June 2022)), Pram (http://pfam.xfam.org/ (accessed on May–June 2022)), and InterPro (http://www.ebi.ac.uk/interpro/ (accessed on May–June 2022)) databases. In total, 304 proteins from Archaea taxa were analysed (TACK Group, Candidatus Thermoplasmatota, DPANN, Asgard Group, and Euryarchaeota); 2511 proteins from bacteria (449 proteins from minor bacteria species, 839 from Proteobacteria (66 minor Proteobacterial species; 178 alphaproteobacteria; 186 betaproteobacteria; 286 gammaproteobacteria; 60 deltaproteobacteria; 63 epsilonproteobacteria); 89 from FCB group, 1134 from Terrabacteria group); 1823 Eukaryotes (including 145 proteins from minor Eukaryotes (Cryptophyceae, Palpitomonas, Apusozoa, Rhodophyta, Metamonada, Haptista, Discoba, and Amoebozoa), 109 from SAR, 1166 from Viridiplantae and 403 from Opisthokonta. The presence of the membrane transport protein domain was defined with the CD Batch search tool (https://www.ncbi.nlm.nih.gov/Structure/bwrpsb/bwrpsb.cgi (accessed on May–June 2022)), and transmembrane domains were defined with CCTOP tool (http://cctop.ttk.hu/ (accessed on May–June 2022)) [29]. Fragmented and partial proteins were removed from the dataset. The basic domain architecture for the MTP proteins was marked with 10 (TM) transmembrane domains, where the presence of loops of different lengths was accounted for. Although most of the MTP proteins are single-domain proteins, some taxa (SAR, Metazoan) were usually followed by the class B family of seven transmembrane G-protein-coupled receptors (7tm_classB, cd13952) and DEP (Dishevelled, Egl-10, and Pleckstrin) domain (cd04443 or pfam00610). Further proteins were processed with Uniprot ID.

### 4.2. Alignments and Phylogenetic Analysis

Multiple sequence alignments of MTP protein sequences were performed using MUSCLE [30] with default settings in UGENE software [31]. Substitution models test and phylogeny analysis were carried out using the MEGA 11 software [32]. For the maximum likelihood tree [33] the JTT substitution model [34] was selected, assuming an estimated proportion of invariant sites and four gamma-distributed rate categories to account for rate heterogeneity across sites. The gamma shape parameter was estimated directly from the data. Reliability for the internal branch was assessed using the bootstrapping method (1000 bootstrap replicates). The same settings were used in another tree reconstruction method, neighbour joining [35], with similar results obtained (Appendix A).

Intrinsically disordered regions (IDRs) were identified with the IUPred2 online tool (https://iupred2a.elte.hu/ (accessed on May–June 2022)) [36]. NCBI pBLAST was used with default settings against non-redundant protein sequences (nr) (https://blast.ncbi.nlm.nih.gov/Blast.cgi (accessed on May–June 2022)) [37].

### 4.3. Structure Search, Modelling, Alignment, and Visualisation

Protein structure prediction was conducted with the Robetta server [22] (https://robetta.bakerlab.org/ (accessed on May–June 2022)); the quality of reconstructed structures was evaluated with QMEAN [38] (https://swissmodel.expasy.org/qmean/ (accessed on May–June 2022)). Structural similarities were searched with NCBI VAST tools [39] (https://www.ncbi.nlm.nih.gov/Structure/VAST/vastsearch.html (accessed on May–June 2022)); pdb structures were aligned with iPBA web tool [40] (https://www.dsimb.inserm.fr/dsimb_tools/ipba/index.php (accessed on May–June 2022)) and visualised with PyMol (https://pymol.org/2/ (accessed on May–June 2022)).

## 5. Conclusions

In this manuscript, we provide phylogenetic and structural analysis of the membrane transport protein domain. The results of our phylogenetic study suggested the presence of 13 primary clades and detected several cases of probable gene transfer events between distant taxa. The analysis of transmembrane domain composition demonstrated the absence of conservation in TM domains III, V, and X, whereas other TM domains were conserved through all studied taxa. The presence of cytoplasmic and non-cytoplasmic loops of different lengths and location points to species–specific adaptations to transported substrates, intracellular localization, and potential protein–protein interactions. The comparison of modelled structures demonstrated a high level of conservation between MTP domains from different taxa. Furthermore, the high level of structural similarity between archaeal and bacterial MTPs and Na/H exchangers implies close functional relation and, probably, the similarity of the transported substrate. Further experimental data about the functionality and physiological role of MTPs from the other taxa, including fungi, bacteria, archaea, and animals would help to understand the structure–function relationship and their evolutional significance.

## Figures and Tables

**Figure 1 ijms-23-08094-f001:**
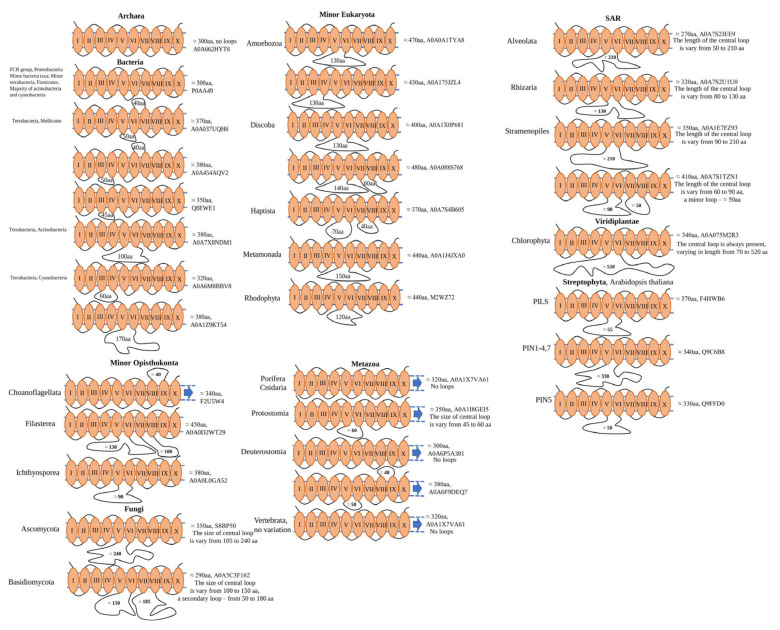
Domain organisation of MTP domain-containing proteins in the different taxa. Designated loop length is applied only to a particular protein (referred to as UniProt ID). Taxa-specific description of loop length is referred to as “no loops” or varies. The presence of other domains is designated with a blue arrow. Upward loops represent non-cytoplasmic, and downward—cytoplasmic loops.

**Figure 2 ijms-23-08094-f002:**
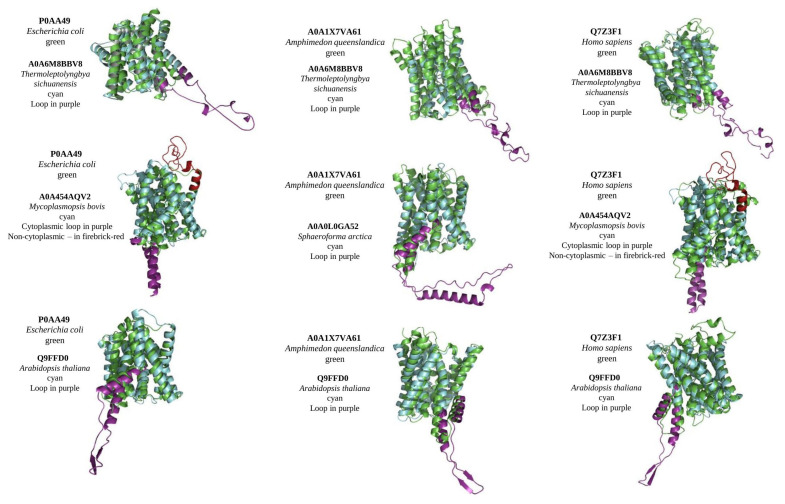
Alignment of structural models of MTP domain-containing proteins from different taxa. Control proteins (green) (with no loops) from bacteria *Escherichia coli*, sponge *Amphimedon queenslandica*, and *Homo sapiens* were aligned with MTPs with loops (cyan) from cyanobacteria *Thermoleptolyngbya sichuanensis* (alignment scores 112, 113, and 112 for every control, respectively), terrabacteria *Mycoplasmopsis bovis* (alignment scores 166 and 169), a unicellular eukaryote *Sphaeroforma arctica* (alignment score 148), and plant *Arabidopsis thaliana* (alignment scores 317, 362, and 289). Non-cytoplasmic loop represented with firebrick-red, cytoplasmic with purple colours.

**Figure 3 ijms-23-08094-f003:**
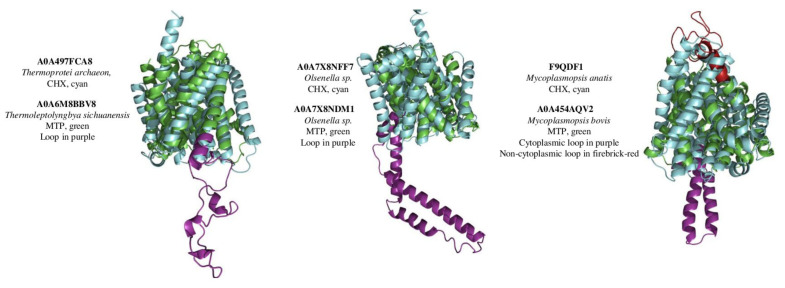
Alignment of structural models of MTP domain-containing proteins with Na/H Exchanger domain. MTPs (green) (with loops) from cyanobacteria *Thermoleptolyngbya sichuanensis* were aligned with Na/H exchanger from archaea *Thermoprotei archaeon* (cyan) (alignment score 146). MTP from *Olsenella* sp. (green) was aligned with Na/H Exchanger from the same bacteria (alignment score 152), and MTP from *Mycoplasmopsis bovis* (green) was aligned with Na/H exchanger from *Mycoplasmopsis anatis* (cyan) (alignment score 221). MTPs’ non-cytoplasmic loop is represented with firebrick-red, cytoplasmic with purple colours.

## Data Availability

All data supporting the findings of this study are available within the paper and within its Appendix A published online.

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
