# Peer review of "Evolution of the Membrane Transport Protein Domain"

_ijms, 2022, doi:10.3390/ijms23158094_

Round 1
Reviewer 1 Report
Here is a comparative analysis of MTP proteins from completely different taxa. The authors did a thorough bioinformatics work and discussed the results well. The manuscript is generally well written, but needs some correction before publication.
The first two paragraphs do not contain a single reference to literary sources.
Lines 9-10, 30-32, 56 and through the whole manuscript: The writing style needs some tweaking to avoid tuffologies and other technical errors. Also, some of the Latin names are in italics, and some are not (for example, lines 106-139). The style needs to be unified. The same goes for the English language. It must be carefully checked.
Lines 28-30, major point (very important): The authors argue that the family of membrane transport proteins is poorly understood to date. However, if you enter "Membrane Transport Proteins" into the PubMed database search bar, more than 576,000 papers come up. The peak of interest in this topic, undoubtedly extremely important, came in 2013, when more than 25,400 articles were published mentioning these keywords. From this moment begins a gradual decline in the number of manuscripts. In this regard, it is extremely difficult to argue that the topic has been little developed, and membrane transport proteins are poorly understood.
Lines 106-108, here and through the whole manuscript: “which is encoding a single-domain protein of approximately 300 amino acids with 10 TM domains” – Here and everywhere in similar places there is an apparent contradiction. First, it talks about a single-domain protein, and then it is specified that it has 10 transmembrane domains. Are these domains or do you mean motives? If there are domains, then the protein is no longer single-domain, but at least 10-domain. It is required to eliminate all such stylistic and semantic contradictions.
Line 109: The closing bracket is missing.
Line 117: Amoebozoa
Lines 149-150: For other taxa where the MTP domain was a part of multi-domain proteins, the sequences of the MTP domain were extracted. – Please point the criteria used for the domain extraction (the start and end of the sequence used for alignment and why this choice was made).
Lines 158-159: “simple Eukaryotic organism Entamoeba histolytica” – I can argue that this organism is simple. Please change the wording.
Lines 160, 161, 167, 169, 179 – Firmicutes
Line 275, Figure 2, major point: This figure is not intuitively understandable. I beg you to make it more clear.
Line 295, Figure 3: I can say almost the same about this figure too. It should be more clearly defined which model corresponds to which particular protein.
Lines 437-449, Conclusion: I think the Conclusion looks like a good Abstract for this article. The Abstract that is now, I do not really like. I would suggest removing the Conclusion altogether, moving the information to an abstract, and expanding it a bit by adding a sentence or two on research methods used in this investigation.
Lines 482-483: The same settings were used in another tree reconstruction method, Neighbour-Joining [35], with similar results obtained (data not shown). – To my opinion, these results in any form should be presented in the Supplement materials section.
Lines 487-493: This paragraph should be separated and numbered as 5.3.
Author Response
Dear Editor and Reviewers,
We greatly appreciate your critical evaluation of our manuscript and helpful comments. Our reply to your comments would be provided point by point, where “A” stands for “Authors”, and “L” for “Lines”, where changes have been implemented. The language of the entire manuscript has been checked and corrected.
____________________________________________________________________________
Here is a comparative analysis of MTP proteins from completely different taxa. The authors did a thorough bioinformatics work and discussed the results well. The manuscript is generally well written, but needs some correction before publication.
The first two paragraphs do not contain a single reference to literary sources.
A: Actually, those two paragraphs are very general, resulting from database and genome/proteome searches. Annotations are mostly automated, so it is very hard to define appropriate paper/s to cite.
Lines 9-10, 30-32, 56 and through the whole manuscript: The writing style needs some tweaking to avoid tuffologies and other technical errors. Also, some of the Latin names are in italics, and some are not (for example, lines 106-139). The style needs to be unified. The same goes for the English language. It must be carefully checked.
A: The Latin names were corrected throughout the entire manuscript (in guidance with the Shenzhen Code, only genus and species names are in italic, and higher taxa (family, order, etc.) are capitalized but not italicized). The language of the entire manuscript has been checked and corrected.
Lines 28-30, major point (very important): The authors argue that the family of membrane transport proteins is poorly understood to date. However, if you enter "Membrane Transport Proteins" into the PubMed database search bar, more than 576,000 papers come up. The peak of interest in this topic, undoubtedly extremely important, came in 2013, when more than 25,400 articles were published mentioning these keywords. From this moment begins a gradual decline in the number of manuscripts. In this regard, it is extremely difficult to argue that the topic has been little developed, and membrane transport proteins are poorly understood.
A: We would like to disagree here. The search criteria "Membrane Transport Proteins" are not very specific. If we will carefully check some of those 576,000 papers, those keywords would be matched for many transporters, not related to the domain of interest. However, the domain-focused search (Pram PF03547 (http://pfam.xfam.org/family/PF03547#tabview=tab0),
or InterPro IPR004776 (https://www.ebi.ac.uk/interpro/entry/InterPro/IPR004776/) would return only a few papers. For example, human GPR155 has only 9 associated papers https://www.uniprot.org/uniprotkb/Q7Z3F1/publications
with more for plant PINs (please, see PIN1 as an example) https://www.uniprot.org/uniprotkb/Q9C6B8/publications
This confusion is coming mostly from the very unprecise title for this domain “Membrane transport protein”. Please, follow the link to the InterPro database link:
(https://www.ebi.ac.uk/interpro/entry/InterPro/IPR004776/) – the very first sentence supports our point:
“This entry represents a mostly uncharacterised family of membrane transport proteins found in eukaryotes, bacteria and archaea.”
Lines 106-108, here and through the whole manuscript: “which is encoding a single-domain protein of approximately 300 amino acids with 10 TM domains” – Here and everywhere in similar places there is an apparent contradiction. First, it talks about a single-domain protein, and then it is specified that it has 10 transmembrane domains. Are these domains or do you mean motives? If there are domains, then the protein is no longer single-domain, but at least 10-domain. It is required to eliminate all such stylistic and semantic contradictions.
A: Indeed, we are sorry for these contradictions. The “Membrane transport protein” refers to the domain as a functional unit, while transmembrane domains – to the structural. Please, see cited manuscripts (plants or animals related) – all of them refer TM as a “domain” ([3][16][20] and many others). We could replace the “TM domain” with “TM region” or “TM segment” but please, keep in mind that the term “TM domain” is widely accepted and used in scientific publications, so such replacement would also be confusing. We have slightly modified text in this sentence (L376).
Line 109: The closing bracket is missing.
A: The bracket was added (L122).
Line 117: Amoebozoa
A: Corrected.
Lines 149-150: For other taxa where the MTP domain was a part of multi-domain proteins, the sequences of the MTP domain were extracted. – Please point the criteria used for the domain extraction (the start and end of the sequence used for alignment and why this choice was made).
A: The domain start/end points were defined with NCBI CD tool (please see 5.1. L501-503).
Lines 158-159: “simple Eukaryotic organism Entamoeba histolytica” – I can argue that this organism is simple. Please change the wording.
A: Corrected.
Lines 160, 161, 167, 169, 179 – Firmicutes
A: Corrected thought the entire manuscript,
Line 275, Figure 2, major point: This figure is not intuitively understandable. I beg you to make it more clear.
Line 295, Figure 3: I can say almost the same about this figure too. It should be more clearly defined which model corresponds to which particular protein.
A: Figure 2 and 3 were modified according to reviewer suggestions.
Lines 437-449, Conclusion: I think the Conclusion looks like a good Abstract for this article. The Abstract that is now, I do not really like. I would suggest removing the Conclusion altogether, moving the information to an abstract, and expanding it a bit by adding a sentence or two on research methods used in this investigation.
A: We appreciate your suggestions. The Abstract and the Conclusion were modified.
Lines 482-483: The same settings were used in another tree reconstruction method, Neighbour-Joining [35], with similar results obtained (data not shown). – To my opinion, these results in any form should be presented in the Supplement materials section.
A: We have additionally added Supplementary figure 5.
Lines 487-493: This paragraph should be separated and numbered as 5.3.
A: The section was modified according to reviewer suggestions (L527-534).
Reviewer 2 Report
Membrane transport proteins (MTPs) are critical to cellular function and prevalent in all Domains of life (Bacteria, Archaea, and Eukarya). In this manuscript, Dabravolski and Isayenkov employ in silico techniques to explore the molecular evolution of MTPs from different phylogenetic taxa. The authors analyzed domain structure/organization of MTPs across diverse taxa using Uniprot and Pfam data and loop structures, as well as modeled ribbon structures of phylogenetically diverse MTP representatives. Phylogenetic analyses suggested 13 primary clades and the possibility of gene transfer events between species of different taxa. The authors suggested that the presence of loops of different lengths points to species-specific adaptations to transported substrates, intracellular localization, and protein-protein interactions. This study lacks empirical data; however, computational analyses and structure modeling based on primary sequence information can help to guide experimental studies on this important group of transmembrane proteins. Overall, the scientific approach is sound and the conclusions drawn from the study are reasonable. The authors need to extensively improve the quality of writing and accuracy prior to publication. Some specific comments are provided below.
(1) Abstract (line 17): "in-deep analysis" should be "in-depth analysis".
(2) Introduction (line 40: "Huntington disease" should be "Huntington's disease".
(3) Results (line 100): provide a more specific, descriptive subheading than "Sequences identification", which should be "Sequence identification". In addition, "Fermicutes" (line 179) is misspelled and should be "Firmicutes". The word "contains" is also misspelled in that same sentence.
(4) In Figure 1, only one archaeal MTP-domain-containing protein is represented. Is this one MTP domain representative of MTPs from members in the two main phyla, the Euryarchaeota and Thermoproteota (formerly, Crenarchaeota)? Please clarify in the text description for Figure 1.
(5) Results (subsection 2.4, line 235): "Loops analysis" should be "Loop analysis". In addition, please do not enumerate the results in Subsection 2.4. Incorporate findings in paragraph form, providing details.
(6) The quality of writing needs to be improved throughout the manuscript for clarity and accuracy.
Author Response
Dear Editor and Reviewers,
We greatly appreciate your critical evaluation of our manuscript and helpful comments. Our reply to your comments would be provided point by point, where “A” stands for “Authors”, and “L” for “Lines”, where changes have been implemented. The language of the entire manuscript has been checked and corrected.
____________________________________________________________________________
Membrane transport proteins (MTPs) are critical to cellular function and prevalent in all Domains of life (Bacteria, Archaea, and Eukarya). In this manuscript, Dabravolski and Isayenkov employ in silico techniques to explore the molecular evolution of MTPs from different phylogenetic taxa. The authors analyzed domain structure/organization of MTPs across diverse taxa using Uniprot and Pfam data and loop structures, as well as modeled ribbon structures of phylogenetically diverse MTP representatives. Phylogenetic analyses suggested 13 primary clades and the possibility of gene transfer events between species of different taxa. The authors suggested that the presence of loops of different lengths points to species-specific adaptations to transported substrates, intracellular localization, and protein-protein interactions. This study lacks empirical data; however, computational analyses and structure modeling based on primary sequence information can help to guide experimental studies on this important group of transmembrane proteins. Overall, the scientific approach is sound and the conclusions drawn from the study are reasonable. The authors need to extensively improve the quality of writing and accuracy prior to publication. Some specific comments are provided below.
(1) Abstract (line 17): "in-deep analysis" should be "in-depth analysis".
A: The entire abstract was modified (L18-29)
(2) Introduction (line 40: "Huntington disease" should be "Huntington's disease".
A: Corrected as suggested (L51 and 55)
(3) Results (line 100): provide a more specific, descriptive subheading than "Sequences identification", which should be "Sequence identification". In addition, "Fermicutes" (line 179) is misspelled and should be "Firmicutes". The word "contains" is also misspelled in that same sentence.
A: The subsection title was modified (L112)
(4) In Figure 1, only one archaeal MTP-domain-containing protein is represented. Is this one MTP domain representative of MTPs from members in the two main phyla, the Euryarchaeota and Thermoproteota (formerly, Crenarchaeota)? Please clarify in the text description for Figure 1.
A: Yes, it is correct. Among 304 analysed archaeal proteins (TACK Group, Candidatus Thermoplasmatota, DPANN, Asgard Group and Euryarchaeota) no loops were detected, therefore, only one representative protein was shown.
The text was modified (L118).
(5) Results (subsection 2.4, line 235): "Loops analysis" should be "Loop analysis". In addition, please do not enumerate the results in Subsection 2.4. Incorporate findings in paragraph form, providing details.
A: We appreciate your suggestions. Section 2. 4 was modified (L253-276).
(6) The quality of writing needs to be improved throughout the manuscript for clarity and accuracy.
A: The language of the entire manuscript has been checked and corrected.
Round 2
Reviewer 1 Report
Dear Colleagues,
Thank you for the changes made in manuscript and reasoned responses to the remarks.